# Comparative Study on Pale, Soft and Exudative (PSE) and Red, Firm and Non-Exudative (RFN) Pork: Protein Changes during Aging and the Differential Protein Expression of the Myofibrillar Fraction at 1 h Postmortem

**DOI:** 10.3390/foods10040733

**Published:** 2021-03-30

**Authors:** Rui Liu, Guo-Yue Wu, Ke-Yue Li, Qing-Feng Ge, Man-Gang Wu, Hai Yu, Sheng-Long Wu, Wen-Bin Bao

**Affiliations:** 1Industrial Engineering Center for Huaiyang Cuisine of Jiangsu Province, College of Food Science and Engineering, Yangzhou University, Yangzhou 225127, China; ruiliu@yzu.edu.cn (R.L.); MX120180904@yzu.edu.cn (G.-Y.W.); MX120190932@yzu.edu.cn (K.-Y.L.); mgwu@yzu.edu.cn (M.-G.W.); yuhai@yzu.edu.cn (H.Y.); 2Jiangsu Key Laboratory of Animal Genetic Breeding and Molecular Design, College of Animal Science and Technology, Yangzhou University, Yangzhou 225009, China; slwu@yzu.edu.cn

**Keywords:** PSE meat, RFN meat, myofibrillar protein fraction, postmortem aging, proteomics

## Abstract

In this paper, the protein changes during aging and the differences in the myofibrillar protein fraction at 1 h postmortem of pale, soft and exudative (PSE), and red, firm and non-exudative (RFN) pork *longissimus thoracis* (LT) were comparatively studied. The PSE and RFN groups were screened out based on the differences in their pH and lightness (*L**) at 1 h, and their purge loss at 24 h postmortem. Based on the measured MFI, desmin degradation, and sodium dodecyl sulfate-polyacrylamide gel electrophoresis (SDS-PAGE) analysis, PSE meat presented more significant changes in the myofibrillar protein fraction compared to RFN meat during postmortem aging. Through liquid chromatograph-mass spectrometer/mass spectrometer (LC-MS/MS) analysis, a total of 172 differential proteins were identified, among which 151 were up-regulated and 21 were down-regulated in the PSE group. The differential proteins were muscle contraction, motor proteins, microfilaments, microtubules, glycolysis, glycogen metabolism, energy metabolism, molecular chaperones, transport, and enzyme proteins. The AMP activated protein kinase (AMPK) signaling pathway, HIF-1 signaling pathway, calcium signaling pathway, and PI3K-Akt signaling pathway were identified as the significant pathways related to meat quality. This study suggested that the different changes of the myofibrillar protein fraction were involved in the biochemical metabolism in postmortem muscle, which may contribute to the molecular understanding of PSE meat formation.

## 1. Introduction

With a reported incidence of 19.17%, pale, soft, and exudative (PSE) meat has always been a major concern in the pork industry [1]. To consumers, PSE meat is less favorable than red, firm, and non-exudative (RFN) meat, as appearance is one of the most important attributes of pork [2]. Moreover, the protein denaturation in PSE meat leads to poor processability and a low yield, resulting in significant losses and a severe hindrance to the pork industry [3,4].

Genetic defects and pre-slaughter stress have been identified as major inducing factors of PSE meat [1,5]. After exsanguination, the myocytes enter into a hypoxic-ischemic environment, accelerating glycolysis and energy metabolism, resulting in lactic acid accumulation and pH decline. Once the pH falls to the isoelectric point of muscle proteins, the gaps between the thick filaments and thin filaments start to narrow, and the space between the myofibrils starts to compress, which limits water retention. As the water loss of the myocytes reaches a certain extent, the filaments become thinner, and the reflectance of incident light declines, resulting in paleness [1]. If exposed to high temperatures, the muscle protein denaturation would accelerate [6], leading to further paleness and lower water holding capacity (WHC) [7].

Myofibrillar proteins, accounting for 50–55% of the total proteins in pig muscle, are responsible for muscle contraction [8]. The integrity of myofibril and the degradation of myofibrillar proteins play a decisive role in meat quality during postmortem aging [9]. Specifically, desmin decomposed from myofibrils is reported to be closely correlated to WHC [10,11]. Studies have shown that the content of intact desmin, dystrophin, and troponin T2 in PSE meat was significantly higher than that in the control group [12,13]. However, it has also been reported that the rapidly decreasing pH in PSE meat promotes the degradation of desmin, titin, and nebulin [14], whereas the relatively high temperature in PSE meat promotes the degradation of troponin-T [15]. This inconsistency might be attributed to the variations in animal species and the pre-slaughter stimulus, which have different postmortem biochemical and physiological impacts.

The molecular-level understanding of PSE meat is limited to glycolysis and pH-induced protein denaturation, which indirectly influence muscle contraction and myofibril integrity during postmortem aging [16]. However, the different protein expression of the myofibrillar fraction post-slaughter and the changes in myofibrillar proteins during postmortem aging of PSE meat have rarely been investigated. Thus, this comparative study was conducted in order to identify the differential protein expression in the myofibrillar fraction at 1 h postmortem through proteomics, and to observe the myofibrillar protein changes during postmortem aging in PSE meat and RFN meat in order to gain a better understanding of the mechanisms affecting the meat quality.

## 2. Materials and Methods

### 2.1. Sample Collection

A total of 16 castrated Duroc × Landrace × Yorkshire crossbred pigs with an average weight of 100 ± 10 kg were electrically stunned and slaughtered at a commercial meat processing plant (Sushi Meat Co., Ltd., Huai’an, Jiangsu, China). The slaughtering process was carried out according to the *Ordinance on Pig Slaughtering Management in China*. The pH of the *longissimus thoracis* (LT) was measured at the third rib in order to screen out eight possible PSE (pH < 5.8) and eight possible RFN (5.8 < pH < 6.0) meats [17] within 1 h postmortem. The primarily-screened LT muscles between the third and thirteenth rib were removed from the right half of each carcass (the third rib was considered as the top side, and the thirteenth rib was considered to be the bottom side). The LT muscle of each carcass was cut into three pieces with approximately equal size along the direction from the top to the bottom, respectively assigning as samples of 1 h, 24 h, and 72 h postmortem. Each piece of meat was marked, and its weight was recorded for further quality measurement and the preparation of biochemical samples. The lightness (*L**) value of the LT muscle at 1 h postmortem sample was detected, and then a portion of 50 g was minced to be frozen in liquid nitrogen and stored at −80 °C for further biochemical and proteomics analysis. Meanwhile, the other two pieces of meat were individually vacuum-packed and stored at 4 °C for pH, *L**, and purge loss measurements at 24 h and 72 h postmortem aging. The meat samples for biochemical analysis at 24 h and 72 h postmortem were also prepared similarly to the above. Finally, four PSE samples and four RFN samples were screened out according to the criteria developed by Chmiel et al. [17] and Warner et al. [18], with a slight modification: PSE, pH < 5.8, *L** > 50 at 1 h postmortem, and purge loss > 5% at 24 h postmortem; RFN, pH ≥ 5.8, *L** ≤ 50 at 1 h postmortem, and purge loss ≤ 5% at 24 h postmortem.

### 2.2. The pH, Color, and Purge Loss

The pH of the meat at 1 h and 24 h postmortem was measured with a portable pH meter (Testo 205, Germany) according to the method described by Lomiwes et al. [19]. The pH meter was calibrated with standardized buffer solutions (BBI Life Sciences, Shanghai, China) before the measurement. The pH of each sample was the average value of four different sites. The color of the LT was measured with a colorimeter (CR-400, Konica Minolta, Japan) at 1 h and 24 h postmortem based on the method developed by Zheng et al. [20], with slight modifications. The light source of the colorimeter was D65. The measuring diameter was 8 mm. The colorimeter was calibrated with a standard white plate each time before measuring. The samples were taken out of the package and wiped dry. The sections of the samples were taken along the muscle fiber, and were exposed to air for 10 min. The *L** value of the meat was measured by averaging the values of three different points of the surface. The purge loss was measured using the method developed by Cardoso et al. [21], with slight modifications. The initial weight of the LT was recorded before vacuum packing. After stored at 4 °C for 24 h and 72 h, the samples were unpacked and wiped dry with 100 × 100 mm filter paper (Taizhou Oak Filter Paper Factory, Taizhou, China) before weighing again. The purge loss was calculated as the percentage of the weight loss over the initial weight of each sample.

### 2.3. The Myofibrillar Fragmentation Index

The myofibrillar fragmentation index (MFI) was measured at 1 h, 24 h, and 72 h postmortem according to the method developed by Qian et al. [22], with slight modifications. In brief, 0.5 g of the muscle sample was homogenized (Ultra Turrox T25 basic, IKA, German) with 5 mL pre-cooled buffer A (100 mM KCl, 20 mM K_2_HPO_4_, 1 mM ethylenediamine tetraacetic acid (EDTA), 1 mM MgCl_2_) in an ice bath three times, for 30 s each time, with 60 s intervals. The homogenate was centrifuged at 3000× *g* for 15 min at 4 °C, and the supernatant was discarded. The pellet was washed with 5 mL pre-cooled buffer A. After that, the pellet was dissolved in 1.25 mL buffer A and then filtered through a 200-mesh nylon screen (Anping Jiufeng Mesh Manufacturing Co. Ltd., Anping, China) in order to remove the connective tissue. The filtrate was centrifuged and resuspended in 1.25 mL buffer A and filtered again. The protein concentration of the final filtrate was measured with the Biuret method [23], and was then adjusted to 0.5 mg/mL in order to measure the absorbance at 540 nm (infinite 200 PRO multimode reader, Tecan, Switzerland). The MFI value was defined as A540 multiplied by 200.

### 2.4. SDS-PAGE and Western Blotting

The myofibrillar protein fractions were prepared with the method developed by Xiong and Brekke [24], with slight modifications. In brief, 0.5 g well-chopped LT sample was homogenized twice at 4 °C with 4.5 mL of the pre-cooled phosphate buffer (0.1 M NaCl, 10 mM Na_2_HPO_4_·12H_2_O, 2 mM MgCl_2_·6H_2_O and 1 mM (ethylene bis (oxyethylenenitrilo) tetra-acetic acid (EGTA) at pH 7.0), for 15 s each time, with a 30 s interval. The homogenate was then centrifuged at 2000× *g* for 15 min at 4 °C. After the centrifugation, the supernatant was slowly decanted. Then, the precipitate was washed twice with 4.5 mL phosphate solution, and three times with 4.5 mL 0.1 M NaCl, according to the steps above. The precipitate of the centrifuged sample was collected as the myofibrillar protein fraction. The protein concentration was measured with a Pierce BCA Protein Assay kit (Thermo Fisher Scientific, IL, USA) and adjusted to 4 mg/mL with HENS buffer (100 mM HEPES, pH 7.8, 1 mM EDTA, 0.1 mM Neocuproine, and 1% sodium dodecyl sulfate (SDS) (*w*/*w*)). One volume of the diluted sample was treated with one volume of 2× loading buffer (100 mM Tris-HCl, 20% glycerol (*w*/*w*), 4% SDS (*w*/*w*), 0.05% bromophenol blue (*w*/*v*), and 5% β-mercaptoethanol (*v*/*v*)). The resulting mixture was heated at 95 °C for 10 min before the SDS-PAGE and Western blot detection of desmin.

The SDS-PAGE and Western blotting were performed using the method developed by Zhang et al. [25], with slight modifications. In brief, 20 μg myofibrillar protein was loaded onto the SDS-PAGE gel, which consisted of 4% stacking gel and 10% separating gel. The gels were run on a Bio-Rad Mini-Protean II electrophoresis unit (Bio-Rad Laboratories, Hercules, CA, USA) at 90 V for 30 min, and then at 120 V until the indicator line reached the bottom of the gel. The gels were stained with colloidal coomassie brilliant blue solution (10% (*v*/*v*) glacial acetic acid and 0.025% (*w*/*v*) coomassie brilliant blue R-250) for 1 h, and were then incubated with the destaining solution (10% (*v*/*v*) methanol and 10% (*v*/*v*) glacial acetic acid) for 24 h. The gels were scanned with the scanner Gel Doc XR+ system (Bio-Rad Laboratories, Hercules, CA, USA). In order to detect desmin, the separating gel of the myofibrillar proteins was transferred to a polyvinylidene fluoride (PVDF) membrane at 90 V, 4 °C for 90 min. The membranes were then blocked with 5% (*w*/*v*) non-fat dry milk in Tris-buffered saline containing Tween-20 (TBST; 20 mM Tris-base, 137 mM NaCl, 5 mM KCl and 0.05% (*v*/*v*) Tween-20) for 2 h at room temperature. Dilutions of 1:2,000 in TBST buffer were used for the primary antibodies (mouse polyclonal anti-desmin antibodies, RD301, Abcam, Cambridge, UK). The membranes were incubated with the primary antibodies overnight at 4 °C. After being washed three times (10 min each time) with TBST buffer, goat-anti-mouse secondary antibodies, at dilutions of 1:80,000, were incubated with the membranes for 90 min at room temperature. After three washes with TBST buffer, the membranes were then stained with BeyoECL Moon buffer (Beyotime Institute of Biotechnology, Haimen, China) for 90 s and manually exposed to X-Rays in a dark room. The X-Ray film was scanned with an Epson perfection V30 SE scanner (Epson, Nagano, Japan).

### 2.5. LC-MS/MS Analysis

Nanoscale liquid chromatography coupled to high-resolution nano-electrospray ionization tandem mass spectrometry (nLC-nESI-HRMS) was used to identify the differential protein expression in the myofibrillar fraction according to the method described by Cox et al. [26], with a slight modification. The myofibrillar protein fraction at 1 h postmortem was obtained from the SDS-PAGE procedure, and the concentration was adjusted to 2.5 mg/mL with HENS buffer. The protein digestion by trypsin was performed with the filter-aided sample preparation (FASP) method. Firstly, 100 μg of protein from each sample was incorporated into 1 M DTT, to a final concentration of 100 mM, and was incubated in boiling water for 5 min. Next, the samples were transferred to 10 kDa ultrafiltration centrifuge tubes. After the addition of 200 µL of 8 M urea in 150 mM Tris buffer pH 8.0 (UA buffer), the protein samples were centrifuged at 12,000× *g* for 15 min, and the filtrate was discarded. Then, the samples were alkylated with 100 µL iodoacetamide (50 mM IAA in UA buffer) before oscillating at 600 rpm for 1 min. After that, the samples were incubated in the dark at room temperature for 30 min, and then centrifuged at 12,000× *g* for 10 min. Another 100 µL UA buffer was added and centrifuged at 12,000× *g* for 10 min; the steps of adding the UA buffer and centrifuging were repeated twice. Subsequently, the protein was further washed with 100 µL 50 mM NH_4_HCO_3_ buffer three times. A dose of 40 µL trypsin buffer (3 µg of trypsin in 40 µL of NH_4_HCO_3_ buffer) was added to each sample, and the mixture was oscillated at 600 rpm for 1 min before incubating for 18 h at 37 °C. Following centrifugation at 12,000× *g* for 10 min, the filtrate was collected. After adding 10 µL 0.1% trifluoroacetic acid (TFA) solution, the digested peptides were desalted on a C18 Cartridge (75 μm × 150 mm, 3 μm, EASY-Spray, Thermo Fisher Scientific, USA), and then vacuum lyophilized. Finally, the peptides were reconstituted with 0.1% formic acid (FA), and the peptide concentration was measured at 280 nm using a NanoDrop One spectrophotometer (Thermo Fisher Scientific, MA, USA) for the LC-MS/MS analysis.

A dose of 2 µg peptides (the concentration of each protein sample was 0.5 μg/μL) was injected into an Easy nLC 1200 system (Thermo Fisher Scientific, RD, USA) at a nanoliter flow rate. The trap column (100 μm × 20 mm, 5 μm, C18, Dr. Maisch GmbH, Ammerbuch, Germany) and analytical column (75 µm × 150 mm, 3 µm, C18, Dr. Maisch GmbH, Ammerbuch, Germany) were used at a flow rate of 300 nL/min for the LC-MS/MS analysis. The columns were equilibrated with 95% solution A (0.1% formic acid aqueous solution) and a linear gradient of solution B (0.1% formic acid acetonitrile (*v*/*v*); 80% acetonitrile (*v*/*v*)) was used to separate the peptides. The procedure was set as follows: B linear gradient from 5% to 8% for 0–2 min, B linear gradient from 8% to 23% for 2–90 min, B linear gradient from 23% to 40% for 90–100 min, B linear gradient from 40% to 100% for 100–108 min, B linear gradient maintained at 100% for 108–120 min. The spray voltage was set to 2 kV.

After the peptide separation, data-dependent acquisition (DDA) mass spectrometry was performed on a Q-Exactive HF-X mass spectrometer (Thermo Fisher Scientific, CA, USA) equipped with nano-electrospray ionization (nESI) (Nanospray Flex Ion Sources, Thermo Fisher Scientific, MA, USA), which was used as a nano interface. DDA is an accelerated and autonomous data acquisition mode [27]. In this mode, the MS instrument performed an MS full-scan (MS1 scan), immediately followed by MS2 analysis on a list of precursor ions selected by their abundances from the full-scan spectrum [27,28]. The analysis lasted for 120 min in the positive ions mode. The MS1 scan was performed, and the parameters were set as follows: scanning range of the parent ion, 300–1800 *m/z*; resolution, 60,000 (*m/z*: 200); automatic gain control (AGC) target, 3 × 10^6^; maximum injection time (IT), 50 ms. The number of microscans was set as 1. The 20 precursor ions with the highest intensity from each MS1 scan were selected for high energy collision dissociation (HCD). A resolution of 15,000 (*m/z*: 200), an AGC target of 1 × 10^5^, a maximum IT of 50 ms, an intensity threshold of 2 × 10^4^, an isolation window of 1.6 *m/z*, and a normalized collision energy of 28 were set in the MS2 scan. The number of microscans was set as 1, and the scanned peptides were dynamically excluded for 30 s. The above procedure of LC-MS/MS was performed in triplicate. The raw files were imported into MaxQuant software (version 1.6.1.0, Max Planck Institute of Biochemistry, Martinsried, Germany) for the protein identification against the database Uniprot-Sus scrofa (pig) (http://geneontology.org/). The enzyme was set as trypsin, and the maximum missed cleavages were set to 2. The main search peptide tolerance, first search peptide tolerance, and MS/MS tolerance were 4.5 ppm, 20 ppm, and 20 ppm, respectively. The peptide spectral matching FDR and the protein FDR were both set as 0.01. Razor and unique peptides were used for the protein quantification.

### 2.6. Bioinformatics Analysis

After the screened proteins were obtained, the protein content with a fold change (FC) > 1.5 or < 0.667 (PSE vs. RFN) and a *p* < 0.05 was identified as significantly differential proteins. The significant enrichment analysis of the Gene Ontology (GO) (http://geneontology.org/, accessed on 21 September 2020) annotation was evaluated, and the level of significance for the protein enrichment of a given GO term was tested by Fisher’s Exact Test. A Kyoto Encyclopedia of Genes and Genomes (KEGG) (http://www.genome.jp/kegg/, accessed on 21 September 2020) pathway enrichment analysis was performed. The significant level of protein enrichment in each pathway by Fisher’s Exact Test in the unit of the KEGG pathway was obtained in order to identify the significant metabolic and signal transduction pathways. The protein–protein interaction (PPI) network was obtained from the Omicsbean system (http://www.omicsbean.cn, accessed on 21 September 2020).

### 2.7. Statistical Analysis

The significant differences between the samples of RFN meat and PSE meat within the same aging time were analyzed by a paired samples T-test with SPSS version 16.0 software (SPSS Inc., Chicago, IL, USA). The significant differences between different aging times within the same group were analyzed by Duncan’s multiple range test, and the results were expressed as mean ± standard error. The level of significance was set at *p* < 0.05. The band densities of all lanes were quantified using Quality One software (Version 4.6.2, Bio-Rad, USA).

## 3. Results

### 3.1. Purge Loss, pH, and Color

The LT muscles were divided into the RFN meat group and the PSE meat group according to the pH at 1 h postmortem, and the *L** and purge loss at 24 h postmortem (Table 1). The pH of the PSE meat was significantly lower than that of the RFN meat group at 1 h and 24 h postmortem (*p* < 0.05). Purge loss is an important indicator of pork WHC. The purge loss of PSE meat at 24 h and 72 h postmortem increased significantly—by more than 5%—compared to the RFN meat (*p* < 0.05). The *L** of the PSE meat was significantly higher than that of RFN meat at 1 h and 24 h postmortem.

### 3.2. MFI and SDS-PAGE

The MFI values of the PSE and RFN meat illustrated in Figure 1a showed that the MFI in the PSE meat was significantly higher than that of the RFN meat after 1, 24, and 72 h of postmortem aging (*p* < 0.05), suggesting a greater diversity of myofibrillar proteins in the PSE meat as compared to the RFN meat. The SDS-PAGE for the myofibrillar protein changes during postmortem aging showed a visible variation of the protein between the two groups (Figure 1b), which was semi-quantified in Table 2. It can be seen that there were more protein bands presented in the PSE meat than in the RFN meat. For example, band 3 and band 8 only appeared in the PSE meat. The relative quantitative intensity of band 1 and band 6 in the RFN meat was higher than that of the PSE meat (*p* < 0.05), while the protein content of band 4 and 7 in the PSE group was higher than that of the RFN group at 1 h and 24 h postmortem (*p* < 0.05). Besides this, the relative quantitative value of bands 1, 2, and 7 in the RFN meat increased significantly with time, which was inconsistent with those in the PSE group (*p* < 0.05). Therefore, the change of the myofibrillar protein fraction was significantly different between the PSE and RFN groups, which possibly contributed to the variation of the meat quality. 

### 3.3. Desmin Degradation

The western blotting detection of desmin is shown in Figure 2. Intact desmin (54 kDa) was present in both the RFN and PSE meat during 72 h of the postmortem aging (Figure 2a). The desmin degradation was significant, because the intact desmin band intensity decreased during the postmortem aging. The degraded desmin in the PSE meat was significantly higher than that in the RFN meat after 72 h of postmortem aging (*p* < 0.05, Figure 2b), suggesting a greater proteolytic potential of the myofibrillar protein fraction in the PSE meat, which coincided with the MFI values.

### 3.4. Myofibrillar Protein Identification and Quantification

Label-free quantitative proteomics was used to determine the significant difference in the myofibrillar protein fraction at 1 h postmortem between the RFN and PSE meat groups. The representative MS base peak chromatograms of the PSE and RFN meat are shown in Figure 3a,b, respectively. A total of 719 proteins were identified (Appendix A), among which 650 proteins were found in both the PSE and RFN meat, 66 proteins were found only in PSE meat, and three proteins were found only in the RFN meat. As shown in Figure 3c,d, of the 172 differential proteins detected (PSE vs. RFN, FC > 1.5 or <0.67, *p* < 0.05, as shown in Appendix A), 151 were up-regulated and 21 were down-regulated in the PSE group, as compared to the RFN group.

The main differential proteins were associated with muscle contraction, motor proteins, microfilaments, microtubules, glycolysis, glycogen metabolism, energy metabolism, molecular chaperones, transport, and enzyme proteins (Table 3). The most observed myofibrillar proteins showed a significant difference between the two groups. The expression of β-actin (fragment), nebulin, and myosin heavy chain (fragment) in the RFN meat was higher than that in the PSE meat, while the cofilin-2, microtubule-associated protein RP/EB family member 2, and the dynein light chain were higher in the PSE group (*p* < 0.05). The proteins responsible for muscle contraction, including voltage-dependent L-type calcium channel subunit alpha and cardiac phospholamban, were abundant in the PSE meat, while troponin-T was more expressed in the RFN meat (*p* < 0.05). The sarcomeric proteins of PDZ and LIM domain protein 3, IF rod domain-containing protein, WD repeat-containing protein 1 were up-regulated, while beta-sarcoglycan was down-regulated in the PSE group as compared to the RFN group. Glycolytic enzymes—including pyruvate kinase, 2-phospho-D-glycerate hydro-lyase, and glyceraldehyde-3-phosphate dehydrogenase—were also detected, all of which participated in the synthesis of pyruvic acid from d-glyceraldehyde 3-phosphate. Pyruvate kinase was more expressed in the RFN meat (*p* < 0.05), while the other three enzymes were more abundant in the PSE meat (*p* < 0.05). Three proteins related to energy metabolism—i.e., cytochrome oxidase subunit 7a1, calcium-transporting ATPase, and adenylosuccinate synthase isozyme 1—were more expressed in the PSE meat (*p* < 0.05). Heat shock protein (HSP) 90-alpha isoform 2 and HSP beta-1 were found in the samples, and the expression in PSE meat was significantly higher than that in the RFN meat (*p* < 0.05). Many types of enzymes—such as ligase, ATPase, phosphatase, kinase, and hydrolase—were identified from the myofibrillar protein fraction, most of which were more expressed in PSE meat, suggesting that the myocytes in PSE meat underwent active biochemical and metabolic activities at the early stage post-slaughter.

### 3.5. GO Functional Annotation of the Differential Proteins

The GO annotations of the differentially-expressed proteins between the PSE and RFN meat, including biological processes (BP), cellular components (CC), and molecular functions (MF), were analyzed (Figure 4a). Of the cellular components, the differentially-expressed proteins were distributed in the cytoplasm (32%), cytoskeleton and contractile fiber (24%), nucleus (14%), cytosol (10%) and organelle of the vesicle (12%), mitochondrion (4%), and endoplasmic reticulum (4%). Those proteins mainly exerted the protein binding of the molecular functions and participated in various biological processes, including single-organism processes, cellular metabolic and catabolic processes, protein folding, muscle structure development, and protein complex subunit organization (Figure 4a).

### 3.6. KEGG Pathway Analysis and PPI Network of the Differential Proteins

The significance levels of the protein enrichment for each KEGG pathway were calculated in order to identify the metabolic and signal transduction pathways for the differential proteins. As shown in Figure 4b, the main metabolic pathways involved in protein enrichment included carbon metabolism, the biosynthesis of amino acids, fatty acid metabolism and degradation, and glycolysis/gluconeogenesis. The significant cellular processes were the regulation of actin cytoskeleton and focal adhesion. Most importantly, many signal transduction pathways related to meat quality were identified, including the AMPK signaling pathway, HIF-1 signaling pathway, calcium signaling pathway, PI3K-Akt signaling pathway, and oxytocin signaling pathway. The conversion from muscle to meat was intricate due to the interactions across different metabolic and signaling pathways. The PPI network of the differential proteins was analyzed among the significant pathways (Figure 4c). Several co-expressed protein clusters were observed, which are mainly associated with the AMPK signaling pathway, the Hippo signaling pathway, oocyte meiosis, carbon metabolism, and the oxytocin signaling pathway. The protein cluster related to the AMPK signaling pathway was comprised of tr-type G domain-containing protein (EEF2), calcium/calmodulin-dependent protein kinase (CAMK2G), and serine/threonine-protein phosphatase 2A 65 kDa regulatory subunit A alpha isoform (PPP2R1A), etc. The proteins in the carbon metabolism-related cluster included glyceraldehyde-3-phosphate dehydrogenase (GAPDH), beta-enolase (ENO3), s-formylglutathione hydrolase (ESD), enoyl-CoA hydratase (HADHA), and aspartate aminotransferase (GOT2), most of which were up-regulated in the PSE meat compared to RFN meat.

## 4. Discussion

During postmortem aging, the energy supply to the myocytes shifted from aerobic metabolism to anaerobic glycolysis. The accumulation of lactic acid would result in a low pH in the PSE meat, which causes paleness due to increased light scattering [29]. In this study, PSE meat had a lower pH, a higher *L**, and a higher purge loss at the early postmortem stage, which is consistent with the reported studies [13,30,31].

Myofibrils are composed of thick filaments and thin filaments, which are contraction units of myocytes. Muscle contraction is caused by the force interaction between the myosin cross-bridge on the thick filament and actin on the thin filament [32]. It is generally believed that the contractile process of myofibrils in rigor mortis leads to an increase in drip loss, while the disruption of the myofibril integrity during aging loosens the myofibril shrinkage to promote WHC [9]. In this study, the changes of the myofibrillar protein fraction during postmortem aging were detected by the MFI, SDS-PAGE, and degradation of desmin. The MFI reflects the sensitivity of the myofibrils to fragmentation during homogenization [33]. Due to the fracture of the Z-line, the MFI generally increases during postmortem aging. Our results suggested that the myofibrils in PSE meat were prone to disruption. This was consistent with the report from Wilhelm et al. [34], which showed that the MFI of PSE-like meat in broilers was significantly higher than that of the control samples at 24 h postmortem [34]. Desmin is an intermediate filament protein which connects myofibrils to adjacent myofibrils and cell membranes [35]. The degradation of desmin impacts the myofibril integrity and the water permeability of the cell membrane [36]. This study showed that the intact desmin of PSE meat was relatively lower at 72 h postmortem, suggesting a more intense degradation than that in the RFN meat. Coincidentally, Van de Wiel and Zhang [14] reported that desmin was likely a marker protein to predict drip loss, and a high desmin level in muscle corresponded to low drip loss. This inconsistency may be attributed to animal species, age, muscle type, and detection techniques. Moreover, the SDS-PAGE gel showed different changes in the myofibrillar protein fraction of PSE and RFN meat, possibly due to the different extent of the proteolysis and protein denaturation. Both the soluble proteins and structural proteins in the PSE meat were denatured, and myofibrillar degeneration is the main cause of the poor WHC of PSE meat [37]. Therefore, PSE meat exhibited greater changes in its myofibrillar protein fraction during postmortem aging, which may be involved in the formation of PSE meat.

The sensitivity of myocyte to pre-slaughter stress is of great significance to the conversion from muscle to meat, as it affects the biochemical changes of proteins. Actin and myosin are the main components of the myocyte cytoskeleton that maintain the structural integrity of muscles. The presence of actin and myosin fragments may be a potential sensitive index for the early denaturation of muscle protein. The degeneration of myosin leads to the contraction of the myosin head, which pulls the thick and thin filaments tightly together [38]. This contraction causes more liquid discharge from the fiber bundles in addition to the water loss from myofilament contraction caused by the low pH of PSE meat [16,38]. Cofilin-2 participates in the regulation of actin dynamics in the sarcomere by splitting up actin filaments [39]. A lower amount of cofilin-2 results in reduced actin filament depolymerization [40]. As is consistent with previous studies, cofilin-2 was found to be highly expressed in PSE meat, suggesting that it increases the depolymerization of actin filaments, and thus increases the MFI. Nebulin is the main component of gap filament. The degradation of nebulin was pH-dependent, which was maximal at pH 7.0 and minimal at 6.0–6.3 [41]. This study showed that nebulin was low in PSE meat. Warner et al. [18] found that nebulin degraded more in PSE meat. The degradation of nebulin would contribute to the myofibril fragility in the I-band region and the region close to the Z-disk [42], and would in turn affect the myofibril integrity and WHC [22].

Apart from myofibrillar proteins, many sarcoplasmic proteins related to glycolysis, energy metabolism, and molecular chaperones were also detected in the myofibrillar protein fraction of both groups. As is consistent with many reports, the enzymes involved in glycolysis and energy metabolism, such as creatine kinase, glyceraldehyde-3-phosphate dehydrogenase, and adenylate kinase isoenzyme 1 were more expressed in the PSE group, indicating more active metabolic activity at the early stage of postmortem aging [43]. HSPs are expressed in order to protect myocytes and resist apoptosis under stress conditions [44]. The HSPs detected in myofibrils served as molecular chaperones transferred from the sarcoplasmic reticulum to the myocytes upon stress [45]. The expression of HSP 90-alpha isoform 2 and HSP beta-1 was found to be higher in PSE meat in the present study. Zhang et al. [44] reported that HSP 90 was significantly higher in the muscles of the high pH group than that of the low pH group, and the expression of HSP 90 was positively correlated with WHC. The muscle membrane integrity is considered to be related to the postmortem loss of intracellular water [46]. The low pH in muscles reduces not only the HSP 90 level but also the binding ability of HSP 90 to the membrane, leading to a high drip loss [47]. Moreover, the HSP 90 level was reported to be negatively correlated with lightness [47], which is consistent with the higher *L** value in PSE meat. HSP β-1 (HSP27) is a small HSP involved in the regulation of microfilament dynamics by inhibiting the actin polymerization, thus limiting the ability of microfilament formation. Small shock proteins stabilize and prevent protein aggregation during stress [48]. Because HSPs are the most induced proteins during the cellular stress responses of mammalian cells, it is speculated that PSE meat has a higher degree of cellular stress [49]. Under stress, HSPs are transferred from sarcoplasm to the denatured protein, and are thus highly expressed in the myofibrillar protein fraction of PSE meat.

Differentially-expressed proteins were also distributed in the mitochondria, which is reported to affect the meat color and the stability of the meat color by reducing the oxygen partial pressure and myoglobin. Mitochondria also affect the redox state of myoglobin [50], while the content and chemical state of myoglobin mainly determine the color of meat [2]. In this study, the myoglobin in the myofibrillar protein fraction of PSE meat was higher than that in RFN meat. It was speculated that the myoglobin denaturation caused by low pH resulted in the decreased solubility and increased sarcoplasmic protein precipitation [51]. Thus, more myoglobin was detected in the myofibrillar protein fraction of PSE meat.

According to the KEGG signaling pathway enrichment analysis, the up-regulated proteins participated in the biochemical processes of postmortem muscle through the HIF-1 signaling pathway, calcium signaling pathway, AMPK signaling pathway, and the regulation of actin cytoskeleton. The HIF-1 signaling pathway is sensitive to hypoxia. It mediates the genes involved in cell survival and glycolysis in the case of hypoxia, thereby inhibiting apoptosis [52,53]. It is speculated that PSE meat consumes more oxygen in the early postmortem or pre-slaughter period, which activates the HIF signaling pathway to increase glycolysis rate, which is also consistent with the generally-accepted cause of PSE meat. The AMPK signaling pathway is activated by skeletal muscle contraction, and it regulates energy metabolism [54]. This study showed that the AMPKBI domain-containing protein and 5-AMP-activated protein kinase subunit gamma-3 involved in the AMPK signaling pathway were more abundant in the PSE group. Studies have shown that the depletion of pre-slaughter ATP leads to an earlier activation of AMPK, enhancing glycolysis and lactic acid accumulation in the muscle, which harms the meat quality [55]. Calcium-transporting ATPase and calcium/calmodulin-dependent protein kinase participate in the calcium signaling pathway and play a role in regulating skeletal muscle contraction [56]. Wang et al. [31] reported that PSE meat possesses a significantly higher sarcoplasmic calcium concentration than RFN meat after 1 h postmortem aging. With up-regulated proteins and activated calcium signaling pathways, intense muscle contractions could be induced in PSE meat, which impacts the subsequent metabolic processes. The down-regulated proteins participated in the biochemical processes through the PI3K-Akt signaling pathway and apoptosis. Studies have shown that the PI3K-Akt signaling pathway regulates a variety of cell functions and resists ischemia and hypoxia by activating survival-related proteins [57]. A PI3K-Akt signaling pathway is also reported to inhibit ROS-induced apoptosis and inactivate apoptosis-related proteins such as Bcl-2 family members, mammalian target of rapamycin (mTOR) and glycogen synthase kinase-3 [57]. A study has found that apoptosis is related to the formation of PSE meat. The expression of pro-apoptotic factor Bax, cytochrome c, and the activity of caspase 3 increased in PSE meat, whereas the expression of anti-apoptotic factor Bcl-2 decreased [30]. However, the apoptosis in PSE meat still needs further investigation.

## 5. Conclusions

In summary, the primary outcome of this study was that PSE meat presented more proteolytic potential and protein changes of the myofibrillar fraction during postmortem aging, with many up-regulated proteins at 1 h postmortem compared to that of RFN meat. It was suggested that intense metabolic reactions would occur in PSE meat at the early stage of aging by the evidence of significant pathways in differential protein enrichment, including carbon metabolism, biosynthesis of amino acids, fatty acid metabolism and degradation, and glycolysis/gluconeogenesis. Other than the widely-recognized glycolysis and energy metabolism, our findings revealed many other biochemical processes related to meat quality, such as the HIF-1 signaling pathway, the calcium signaling pathway, apoptosis, and the regulation of actin cytoskeleton. This indicates that the different changes of the myofibrillar protein fraction were involved in the biochemical metabolism in postmortem muscle, which enabled us to elucidate the molecular mechanism of PSE meat formation. The biochemistry of postmortem meat is complex due to the intricate pathways and interactions across different metabolic processes. Therefore, the role of the identified signaling pathways in the development of PSE meat should be further investigated.

## Figures and Tables

**Figure 1 foods-10-00733-f001:**
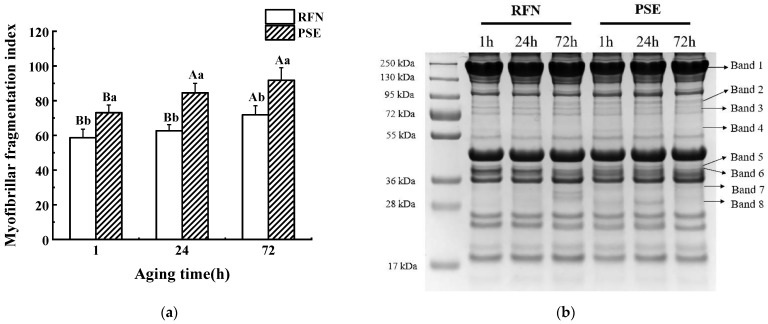
(**a**) Myofibrillar fragmentation index (MFI) value at 1 h, 24 h, and 72 h postmortem of pale, soft and exudative (PSE), and red, firm and non-exudative (RFN) meat; (**b**) representative image of the SDS-PAGE of the PSE and RFN samples during postmortem aging. ^a,b^ Values within the same aging time between different groups are significantly different (*p* < 0.05). ^A,B^ Values within the same group at different aging times are significantly different (*p* < 0.05).

**Figure 2 foods-10-00733-f002:**
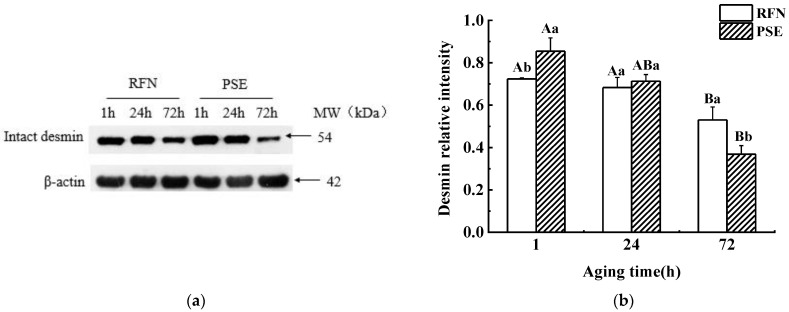
(**a**) Intact desmin contents at 1 h, 24 h, and 72 h postmortem of red, firm and non-exudative (RFN), and pale, soft and exudative (PSE) meat; (**b**) relative intensity of intact desmin in the RFN and PSE meat at 1, 24, and 72 h postmortem. Mean value ± standard error (*n* = 4). ^a,b^ Values within the same aging time between the different groups are significantly different (*p* < 0.05). ^A,B^ Values within the same group at different aging times are significantly different (*p* < 0.05).

**Figure 3 foods-10-00733-f003:**
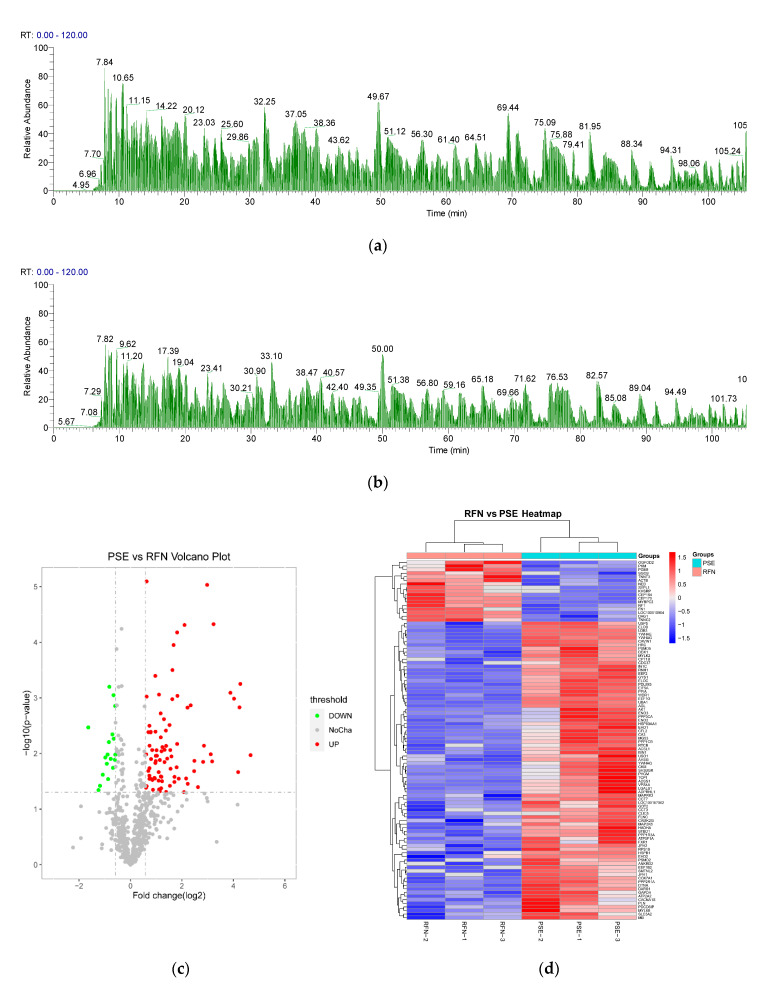
(**a**) Representative mass spectrometry (MS) base peak chromatogram of the myofibrillar protein extracted from RFN meat by nLC-nESI-HRMS; (**b**) representatively MS base peak chromatogram of the myofibrillar protein extracted from PSE meat by nLC-nESI-HRMS; (**c**) comparison of the differential proteins between the pale, soft and exudative (PSE), and red, firm and non-exudative (RFN) groups, as shown in the volcano plot; (**d**) heatmap with the difference FC > 1.5 or < 0.667 and a *p* < 0.05.

**Figure 4 foods-10-00733-f004:**
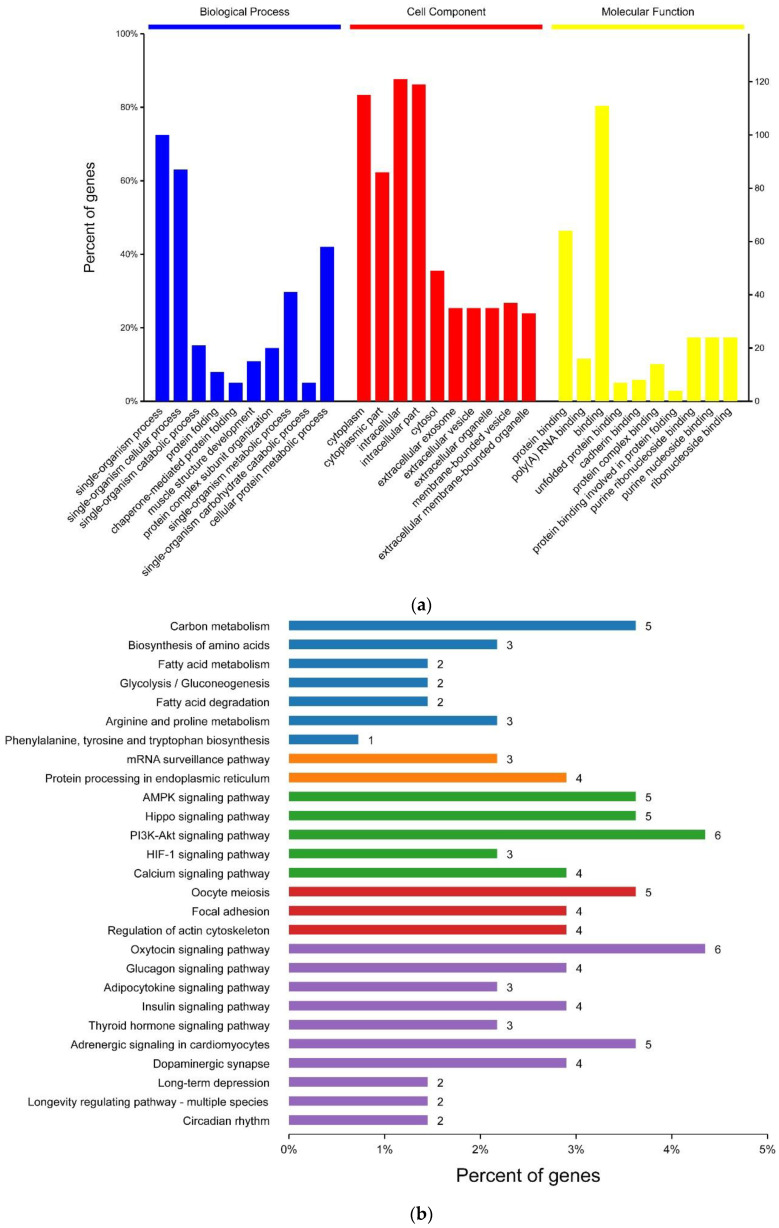
(**a**) GO functional classification; (**b**) significant Kyoto Encyclopedia of Genes and Genomes (KEGG) pathway; (**c**) protein–protein interaction (PPI) network of the differentially expressed proteins between the pale, soft and exudative (PSE), and red, firm and non-exudative (RFN) meat.

**Table 1 foods-10-00733-t001:** The pH, lightness (*L**) and purge loss of red, firm and non-exudative (RFN), and pale, soft and exudative (PSE) meat during postmortem aging.

Index	Postmortem Time(h)	RFN	PSE
pH	1	5.88 ± 0.33 ^a^	5.48 ± 0.16 ^b^
	24	5.63 ± 0.26 ^a^	5.51 ± 0.29 ^b^
*L**	1	42.64 ± 0.93 ^b^	53.27 ± 2.24 ^a^
	24	46.85 ± 0.67 ^b^	56.06 ± 2.12 ^a^
Purge loss(%)	24	1.60 ± 0.44 ^b^	6.96 ± 1.99 ^a^
	72	2.84 ± 0.72 ^b^	8.12 ± 3.46 ^a^

RFN, red, firm, and non-exudative; PSE, pale, soft, and exudative; *L**, lightness. Mean value ± standard error (*n* = 4). ^a,b^ Values within the same row with different superscripts are significantly different (*p* < 0.05).

**Table 2 foods-10-00733-t002:** Band intensities of representative myofibrillar protein among the pale, soft and exudative (PSE), and red, firm and non-exudative (RFN) groups during the postmortem aging.

	RFN			PSE		
	1	24	72	1	24	72
Band 1	1.08 ± 0.01 ^C,a^	1.13 ± 0.03 ^B,a^	1.42 ± 0.02 ^A,a^	1.00 ± 0.00 ^B,b^	1.04 ± 0.01 ^A,b^	1.03 ± 0.01 ^A,b^
Band 2	0.92 ± 0.01 ^C,a^	0.95 ± 0.01 ^B,a^	1.00 ± 0.02 ^A,a^	1.00 ± 0.00 ^B,a^	1.08 ± 0.01 ^A,a^	0.96 ± 0.01 ^C,a^
Band 3	ND	ND	ND	1.00 ± 0.00 ^A^	0.99 ± 0.01 ^B^	0.91 ± 0.01 ^C^
Band 4	0.91 ± 0.01 ^A,b^	0.96 ± 0.01 ^A,b^	0.91 ± 0.01 ^A,a^	1.00 ± 0.00 ^A,a^	0.98 ± 0.01 ^B,a^	0.87 ± 0.01 ^C,b^
Band 5	0.99 ± 0.04 ^AB,a^	1.19 ± 0.15 ^A,a^	0.79 ± 0.01 ^B,a^	1.00 ± 0.00 ^A,a^	1.04 ± 0.03 ^A,a^	0.89 ± 0.03 ^B,a^
Band 6	2.37 ± 0.08 ^A,a^	2.57 ± 0.23 ^A,a^	0.88 ± 0.04 ^B,a^	0.99 ± 0.00 ^A,b^	1.02 ± 0.08 ^A,b^	0.95 ± 0.04 ^A,a^
Band 7	0.85 ± 0.01 ^C,b^	0.93 ± 0.01 ^B,b^	0.98 ± 0.01 ^A,a^	1.00 ± 0.00 ^A,a^	0.97 ± 0.01 ^B,a^	0.96 ± 0.01 ^B,b^
Band 8	ND	ND	ND	1.00 ± 0.00 ^B^	1.07 ± 0.02 ^A^	0.85 ± 0.01 ^C^

RFN, red, firm, and non-exudative; PSE, pale, soft, and exudative. Mean value ± standard error (*n* = 4). ^a–b^ Values within the same aging time between the different groups are significantly different (*p* < 0.05). ^A–C^ Values within the same group at different aging times are significantly different (*p* < 0.05).

**Table 3 foods-10-00733-t003:** Representative differential proteins of pork the myofibrillar fraction between the pale, soft and exudative (PSE), and red, firm and non-exudative (RFN) meat, as identified by LC-MS/MS.

No.	Protein Names	Protein UniProt/ NCBI Accession	Gene Names	Score	FC	*p*-Value
Microfilaments						
1	Beta actin (Fragment)	Q00P29	ACTB	3.97	0.64	<0.01
2	Cofilin-2	Q5G6V9	CFL2	40.98	9.02	<0.05
3	Nebulin	A0A287B9W0	NEB	17.18	0.44	<0.05
Microtubules						
1	Microtubule-associated protein RP/EB family member 2	A0A4X1VCG5	MAPRE2	15.03	1.74	<0.05
Muscle contraction						
1	Troponin T	A0A4X1U8T2	TNNT3	323.31	0.59	<0.05
2	Voltage-dependent L-type calcium channel subunit alpha	A0A5G2QSL4	CACNA1S	43.95	1.55	<0.05
3	Cardiac phospholamban	P61013	PLN	3.11	3.26	<0.05
Motor proteins						
1	Myosin heavy chain (Fragment)	Q95249	/	6.87	0.54	<0.05
2	Dynein light chain	D9U8D1	DYNLL1	16.91	*	*
Other sarcomeric proteins						
1	PDZ and LIM domain protein 3	Q6QGC0	PDLIM3	305.84	3.08	<0.01
2	Beta-sarcoglycan	F1SE70	SGCB	20.01	0.55	<0.05
3	IF rod domain-containing protein	A0A4X1UHN0	/	29.05	2.13	<0.05
4	WD repeat-containing protein 1	K9IVR7	WDR1	177.78	5.06	<0.01
Glycogen metabolism						
1	Glycogen synthase	A0A5G2QKR0	GYS1	67.92	2.46	<0.01
2	4-alpha-glucanotransferase	F1S557	AGL	128.07	4.29	<0.01
Glycolysis						
1	Pyruvate kinase	A0A480JGH8	PKM	49.58	0.42	<0.05
2	2-phospho-D-glycerate hydro-lyase	A0A4X1UZ92	ENO2	5.12	2.29	<0.05
3	Glyceraldehyde-3-phosphate dehydrogenase	P00355	GAPDH	214.00	1.72	<0.05
4	Creatine kinase	A0A5G2QZN6	CKM	323.31	4.68	<0.05
Energy metabolism						
1	Cytochrome c oxidase subunit 7A1	Q8SPJ9	COX7A1	3.38	1.67	<0.01
2	Calcium-transporting ATPase	A0A480TDT7	ATP2A2	295.12	2.50	<0.05
3	Adenylosuccinate synthetase isozyme 1	A0A287BAF3	ADSS1	105.07	7.07	<0.05
Transport						
1	Eukaryotic translation initiation factor 5A	A0A4X1V2D0	EIF5A	98.96	19.27	<0.01
Molecular chaperones						
1	Heat shock protein HSP 90-alpha isoform 2	A0A481CXT9	HSP90AA1	256.08	3.02	<0.05
2	Heat shock protein beta-1	Q5S1U1	HSPB1	323.31	1.67	<0.05
3	Hsp90 chaperone protein kinase-targeting subunit	F6Q4F9	CDC37	10.90	2.26	<0.05
4	T-complex protein 1 subunit	A0A5G2RH19	CCT7	11.29	1.62	<0.01
5	T-complex protein 1 subunit gamma	A0A480WDC3	CCT3	27.42	2.31	<0.01
Enzyme proteins						
1	Dual specificity protein phosphatase	I3LCX3	DUSP3	13.52	*	*
2	26S proteasome -ATPase subunit RPT1	A0A480VIW9	PSMC2	11.63	*	*
3	AMPKBI domain-containing protein	K7GPQ2	PRKAB2	3.15	*	*
4	Flavin reductase (NADPH)	I3LQH7	BLVRB	3.95	*	*
5	Phosphoinositide phospholipase C	F1SRY6	PLCD4	28.16	*	*
6	5-AMP-activated protein kinase subunit gamma-3	K7GM96	PRKAG3	2.49	*	*
7	Peptidyl-prolyl cis-trans isomerase	I3LLH5	PIN1	20.52	*	*
8	S-formylglutathione hydrolase	A0A5S6IDI6	ESD	3.83	*	*
9	Ubiquitin carboxyl-terminal hydrolase	A0A4X1VR78	USP14	7.39	*	*
10	Cathepsin B	A0A287BF94	CTSB	7.84	**	**
11	Ubiquitin-activating enzyme E1	A0A480W380	UBA1	51.87	9.38	<0.01
12	Calcium/calmodulin-dependent protein kinase	F6QB46	CAMK2G	49.06	1.66	<0.05
13	Alpha-1,4 glucan phosphorylase	A0A286ZMZ9	PYGM	323.31	18.18	<0.05
14	Adenylate kinase isoenzyme 1	P00571	AK1	187.73	6.16	<0.05
15	ATP synthase subunit alpha	F1RPS8	ATP5F1A	63.03	1.97	<0.05
16	Aspartate aminotransferase	A0A4X1UT32	GOT2	17.00	1.62	<0.05
17	Striated muscle preferentially expressed protein kinase	A0A481BRQ9	/	64.82	1.53	<0.05
18	26S proteasome non-ATPase regulatory subunit 2	I3LEW5	PSMD2	14.26	2.89	<0.05
19	Aspartate-tRNA ligase, cytoplasmic	A0A480NFZ5	DARS1	29.47	1.54	<0.01
20	Myosin light chain kinase 2	F1S7H3	MYLK2	177.41	1.96	<0.01
21	Vesicle-fusing ATPase	A0A4X1SF26	VPS4A	16.74	2.02	<0.01
22	Protein kinase domain-containing protein	A0A5K1UKA4	MAP2K6	23.01	1.91	<0.05
23	Serine/threonine-protein phosphatase	A0A286ZXJ6	PPP3CA	7.29	2.61	<0.05
24	Serine/threonine-protein phosphatase PP1-beta catalytic subunit	P61292	PPP1CB	26.24	7.24	<0.01
25	Ubiquitin carboxyl-terminal hydrolase	A0A5G2R3H5	USP5	27.48	2.61	<0.01
Others						
1	Peptidyl-prolyl cis-trans isomerase A	P62936	PPIA	12.36	18.84	<0.01
2	Myoglobin	P02189	MB	97.32	2.21	<0.05
3	Alpha-dystroglycan	A0A286ZY59	DAG1	26.12	0.56	<0.01
4	Carbonic anhydrase 3	Q5S1S4	CA3	323.31	25.40	<0.05

FC, fold change. The accession was obtained in Uniprot/NCBI. * means that it was only detected in PSE meat; ** means that it was only detected in RFN meat.

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
