# Peer review of "Comparative Study on Pale, Soft and Exudative (PSE) and Red, Firm and Non-Exudative (RFN) Pork: Protein Changes during Aging and the Differential Protein Expression of the Myofibrillar Fraction at 1 h Postmortem"

_foods, 2021, doi:10.3390/foods10040733_

Round 1
Reviewer 1 Report
Considering the specific topic of the present manuscript, it is opinion of the reviewer that it should be submitted into a more specific journal (e.g. Meat). Moreover, it is not clear to me the final goal of the authors.
Other comments
- line numbers must be inserted to facilitate the reviewer(s) job.
- English has to be improved. Although I am not qualify, I found the manuscript hard to be read and followed.
- There are several scientific terms that are not English, such as "...was chromatographed". Although it can make sense, this is not English, or not yet.
- The analytical part (LC-MS) should be better addressed and it presents several lacks. There is not optimization of the LC method, not information about repeatability, limit of detection, matrix effect etc. No information about the concentration of the analytes and the volume of injection. No enough information about the LC system and the analytical parameters applied, etc
- Data dipendent aquisition (DDA) should be better expained, considering that not all the scientific readers of Foods are familiar with this MS approach
- The accuracy of the method identification should be reported
- A LC-MS chromatogram should be showed in the text
- Considering the the authors applied 0.3 uL/min as LC flow-rate, a nano interface is needed and the hyphenation between the LC system and the MS through the nano-interface should be explained, but no information about the MS interface is reported.The authors should keep in mind that a generic scientific reader should be able to reproduce in toto the scientific experiment basing on the information reported in the manuscript.
Author Response
Dear reviewer:
Thank you very much for the valuable comments concerning our manuscript. We have addressed all the comments as shown in the revised manuscript which we hope will now meet with your approval. Revised portions of the manuscript were tracked.
1. Considering the specific topic of the present manuscript, it is opinion of the reviewer that it should be submitted into a more specific journal (e.g. Meat). Moreover, it is not clear to me the final goal of the authors.
Response: Thank you very much for your kind advice. Meat has comprised an important part of the human foods [1]. There has been a considerable increase (62%) in meat consumption worldwide, with the biggest increase in developing countries (a threefold increase) [1]. Future consumption of animal-based products in developing countries is projected to increase from 29% to 35% in 2030 and 37% in 2050 which compares to an average of 48% in industrial countries [2]. In the USA and the UK, the most important meat sources are from pigs, and meat products such as sausages, burgers, pork pies, etc., account for almost half of all meat consumed in developed countries [1]. Pale, soft, and exudative (PSE) meat has always been a major concern in the pork industry with a reported incidence of 19.17% [3].To the consumers, PSE meat is less favorable than red, firm, and non-exudative (RFN) meat as the appearance is one of the most important attributes of pork [4]. Moreover, the protein denaturation in PSE meat leads to poor processability and low yield, resulting in significant losses and a severe hindrance to the pork industry [5,6]. The molecular level understanding of PSE meat is limited to the glycolysis and pH-induced protein denaturation, which indirectly influence muscle contraction and myofibril integrity post-slaughter [7]. However, the different myofibrillar protein fraction and the changes in myofibrillar proteins during postmortem aging of PSE meat was rarely investigated. Thus, our study was conducted to identify the different myofibrillar protein fraction through proteomics and observe the changes in myofibrillar proteins during postmortem aging in PSE meat and RFN meat to gain a better understanding of the mechanism affecting meat quality. Therefore, this manuscript was submitted to the “Meat” section of “Foods” journal. It is of great appreciation to reviewer for your valuable time to consider this manuscript.
- line numbers must be inserted to facilitate the reviewer(s) job.
Response: We are sorry for the inconvenience for the line number missing and we have inserted line numbers as suggested in the revised manuscript.
- English has to be improved. Although I am not qualify, I found the manuscript hard to be read and followed.
Response: Thanks for your suggestion. The English of this article has been polished by Qingdao Essentials Editing and consulted a native English speaking meat expert for the language of the manuscript. The certificate is shown a attached file.
- There are several scientific terms that are not English, such as "...was chromatographed". Although it can make sense, this is not English, or not yet.
Response: We are sorry for the incorrect terms and we have now checked the terms of the manuscript. This sentence was rewords as follows: "A dose of 2 µg enzymolyzed peptide (the concentration of each protein sample was 0.5 μg/μL) was injected into an Easy nLC 1200 chromatographic System (Thermo Fisher Scientific, RD, USA) at a nanoliter flow rate." (Line 181-183)
- The analytical part (LC-MS) should be better addressed and it presents several lacks. There is not optimization of the LC method, not information about repeatability, limit of detection, matrix effect No information about the concentration of the analytes and the volume of injection. No enough information about the LC system and the analytical parameters applied, etc.
Response: Thanks for your suggestion and we are sorry for the mis-understanding of LC method description. The nanoscale liquid chromatography coupled to high-resolution nano-electrospray ionization tandem mass spectrometry (nano LC–MS/MS) peptide analysis was performed to explore the differential expression protein in myofibrillar fraction between PSE and RFN in the current study. Tandem mass spectrometry (MS/MS) coupled with multidimensional liquid chromatography (LC) together with database searching has emerged as a robust technique of protein identification and characterization [8]. This proteomics mothed was established for decade and employed in many studies [9-14], which allows the highly sensitive identification of hundreds of distinct proteins in a given biomedical sample [15]. Nano LC–MS/MS has become an essential tool in the field of proteomics because of its higher sensitivity than conventional LC–MS/MS [16]. The workflow of the LC-MS-based protein identification was briefly shown as follows. Firstly, the protein extract was digested into peptide fragments by protease. Then the peptide fragment mixture was separated by LC, and the peptide fragments were fragmented to product ions by mass spectrometry. Subsequently, datasets of the product ion spectra are subsequently searched against primary structure databases using search engines to identify the proteins contained in the analytes [17].
Fig. 1. Schematic representation of the approach used in quantitative proteomics studies [18].
In the current study, possibly differential proteins in the myofibrillar fraction between PSE and RFN group were intended to be detected, rather than one or more specific proteins or polypeptides. The detecting procedure of LC–MS/MS method (MaxLFQ) was referring to Cox’s [19], and the technique detection was performed in triplicate to ensure the repeatability. MaxLFQ is a generic label-free quantification technology, was used for the intensity determination and normalization procedure and was fully compatible with any peptide or protein separation prior to LC-MS analysis [19]. Furthermore, MaxLFQ extracted the maximum ratio information from peptide signals in arbitrary numbers of samples to achieve the highest possible accuracy of quantification [19]. For the limit of detection, the mass spectrometric database retrieval software was MaxQuant 1.6.1.0. The enzyme was set as trypsin, and max missed cleavages was set to 2. The main search peptide tolerance, first search peptide tolerance and MS/MS tolerance were 4.5 ppm, 20 ppm and 20 ppm, respectively. Both the peptide spectral matching (PSM) and the protein FDR were set as 0.01 (Line 204-209). In addition, we have added the information about the concentration of the analytes and the volume of injection as follows: "the concentration of each sample before enzymolysis was 2.5 mg/mL, and peptide concentration before injection was 0.5 μg/μL" (Line 159-160, Line181-182). Thus, the LC method description was rewritten by adding more detailed information as you suggested in Line 159-160, 181-182, 204-209.
- Data dipendent aquisition (DDA) should be better explained, considering that not all the scientific readers of Foods are familiar with this MS approach.
Response: Thanks for your suggestion. We have added the sentence as follows: "Data dependent acquisition (DDA) is a common data acquisition strategy, which selects precursor ions for fragmentation based on their abundances [20] ".(Line 191-192) In the label-free proteomics, biological samples are acquired in a data-dependent (DDA) manner, with peptide signals recorded in an intact (MS1) and fragmented (MS2) form [21]. The most intense precursor ions recorded in MS1 are selected at a time for fragmentation by high-energy collision-induced dissociation (HCD) [22,23] and the most abundant precursor ions are selected for tandem mass spectrometry (MS/MS) analysis.
- The accuracy of the method identification should be reported
Response: The mass accuracy of Q-Exactive HF-X Mass Spectrometry (Thermo Fisher Scientific, RD, USA) was less than 3 ppm.
- A LC-MS chromatogram should be showed in the text
Response: As you suggested, the representative LC-MS chromatogram was added in our text. (Line 316 Figure 3.(a), (b))
- Considering the the authors applied 0.3 uL/min as LC flow-rate, a nano interface is needed and the hyphenation between the LC system and the MS through the nano-interface should be explained, but no information about the MS interface is reported.
Response: Thanks for your suggestion. Actually, the Q Exactive Mass Spectrometry equipped with nano-electrospray ionization (nESI) (Nanospray Flex Ion Sources, Thermo Fisher Scientific, USA) was used as the interface [24] (Line193-195). The key feature of the nESI is the reduction in sample flow rate from mL/min of the standard ESI interface to nL/min, which ultimately improves ionization efficiency and sensitivity [15,16]. In the electrospray process, a high voltage (typically 3-5 kV) is applied to the capillary outlet of the spray tip to generate a high potential, and a fine spray of ionized droplets are generated with the aid of heated gas [25]. The gas is also used as an aid to evaporate solvent from the droplets, which become smaller and hence the charge density increases until a point of instability is reached when the droplets break into smaller droplets. This process continues until de-solvated ions are produced, which pass into the high vacuum of the mass analyzer through a small opening guided by electrical potential difference [25].
- The authors should keep in mind that a generic scientific reader should be able to reproduce in to the scientific experiment basing on the information reported in the manuscript.
Response: Thank you very much for your kind advice. We have made corresponding supplements to facilitate reader to reproduce the experiment as above.
References
- Kearney, J. Food consumption trends and drivers. Philosophical transactions of the royal society B: biological sciences 2010, 365, 2793-2807.
- Alexandratos, N.; Bruinsma, J.; Boedeker, G.; Schmidhuber, J.; Broca, S.; Shetty, P.; Ottaviani, M.G. World agriculture: Towards 2030/2050. Interim report. Prospects for food, nutrition, agriculture and major commodity groups. 2006.
- Trevisan, L.; Brum, J.S. Incidence of pale, soft and exudative (PSE) pork meat in reason of extrinsic stress factors. Anais da Academia Brasileira de Ciências 2020, 92.
- Karamucki, T.; Jakubowska, M.; Rybarczyk, A.; Gardzielewska, J. The influence of myoglobin on the colour of minced pork loin. Meat science 2013, 94, 234-238.
- Chen, H.; Wang, H.; Qi, J.; Wang, M.; Xu, X.; Zhou, G. Chicken breast quality–normal, pale, soft and exudative (PSE) and woody–influences the functional properties of meat batters. International Journal of Food Science & Technology 2018, 53, 654-664.
- Li, X.; Feng, F.; Gao, R.; Wang, L.; Qian, Y.; Li, C.; Zhou, G. Application of near infrared reflectance (NIR) spectroscopy to identify potential PSE meat. Journal of the Science of Food and Agriculture 2016, 96, 3148-3156.
- Pearce, K.L.; Rosenvold, K.; Andersen, H.J.; Hopkins, D.L. Water distribution and mobility in meat during the conversion of muscle to meat and ageing and the impacts on fresh meat quality attributes—A review. Meat science 2011, 89, 111-124.
- Madda, R.; Lin, S.-C.; Sun, W.-H.; Huang, S.-L. Plasma proteomic analysis of systemic lupus erythematosus patients using liquid chromatography/tandem mass spectrometry with label-free quantification. PeerJ 2018, 6, 4730.
- Yu, Q.; Tian, X.; Shao, L.; Xu, L.; Dai, R.; Li, X. Label-free proteomic strategy to compare the proteome differences between longissimus lumborum and psoas major muscles during early postmortem periods. Food chemistry 2018, 269, 427-435.
- Montowska, M.; Fornal, E. Label-free quantification of meat proteins for evaluation of species composition of processed meat products. Food chemistry 2017, 237, 1092-1100.
- Ma, Y.; Zhu, J.; Chen, S.; Ma, J.; Zhang, X.; Huang, S.; Hu, J.; Yue, T.; Zhang, J.; Wang, P. Low expression of SPARC in gastric cancer-associated fibroblasts leads to stemness transformation and 5-fluorouracil resistance in gastric cancer. Cancer cell international 2019, 19, 1-12.
- Yu, Q.; Wu, W.; Tian, X.; Hou, M.; Dai, R.; Li, X. Unraveling proteome changes of Holstein beef M. semitendinosus and its relationship to meat discoloration during post-mortem storage analyzed by label-free mass spectrometry. Journal of Proteomics 2017, 154, 85-93.
- He, Y.; Huang, H.; Li, L.-H.; Yang, X. Label-free proteomics of tilapia fillets and their relationship with meat texture during post-mortem storage. Food Analytical Methods 2018, 11, 3023-3033.
- Zhou, C.-Y.; Wang, C.; Tang, C.-B.; Dai, C.; Bai, Y.; Yu, X.-B.; Li, C.-B.; Xu, X.-L.; Zhou, G.-H.; Cao, J.-X. Label-free proteomics reveals the mechanism of bitterness and adhesiveness in Jinhua ham. Food chemistry 2019, 297, 125012.
- Bian, Y.; Zheng, R.; Bayer, F.P.; Wong, C.; Chang, Y.-C.; Meng, C.; Zolg, D.P.; Reinecke, M.; Zecha, J.; Wiechmann, S. Robust, reproducible and quantitative analysis of thousands of proteomes by micro-flow LC–MS/MS. Nature communications 2020, 11, 1-12.
- Gaspari, M.; Cuda, G. Nano LC–MS/MS: a robust setup for proteomic analysis. In Nanoproteomics, Springer: 2011; pp. 115-126.
- Kawakami, T.; Tateishi, K.; Yamano, Y.; Ishikawa, T.; Kuroki, K.; Nishimura, T. Protein identification from product ion spectra of peptides validated by correlation between measured and predicted elution times in liquid chromatography/mass spectrometry. Proteomics 2005, 5, 856-864.
- Matros, A.; Kaspar, S.; Witzel, K.; Mock, H.-P. Recent progress in liquid chromatography-based separation and label-free quantitative plant proteomics. Phytochemistry 2011, 72, 963-974.
- Cox, J.; Hein, M.Y.; Luber, C.A.; Paron, I.; Nagaraj, N.; Mann, M. Accurate proteome-wide label-free quantification by delayed normalization and maximal peptide ratio extraction, termed MaxLFQ. Molecular & cellular proteomics 2014, 13, 2513-2526.
- Fernández-Costa, C.; Martínez-Bartolomé, S.; McClatchy, D.B.; Saviola, A.J.; Yu, N.-K.; Yates III, J.R. Impact of the Identification Strategy on the Reproducibility of the DDA and DIA Results. Journal of Proteome Research 2020, 19, 3153-3161.
- Huang, T.; Bruderer, R.; Muntel, J.; Xuan, Y.; Vitek, O.; Reiter, L. Combining precursor and fragment information for improved detection of differential abundance in data independent acquisition. Molecular & Cellular Proteomics 2020, 19, 421-430.
- Zhang, C.; Zuo, T.; Wang, X.; Wang, H.; Hu, Y.; Li, Z.; Li, W.; Jia, L.; Qian, Y.; Yang, W. Integration of data-dependent acquisition (DDA) and data-independent high-definition MSE (HDMSE) for the comprehensive profiling and characterization of multicomponents from Panax japonicus by UHPLC/IM-QTOF-MS. Molecules 2019, 24, 2708.
- Bateman, N.W.; Goulding, S.P.; Shulman, N.J.; Gadok, A.K.; Szumlinski, K.K.; MacCoss, M.J.; Wu, C.C. Maximizing peptide identification events in proteomic workflows using data-dependent acquisition (DDA). Molecular & Cellular Proteomics 2014, 13, 329-338.
- Michalski, A.; Damoc, E.; Hauschild, J.-P.; Lange, O.; Wieghaus, A.; Makarov, A.; Nagaraj, N.; Cox, J.; Mann, M.; Horning, S. Mass spectrometry-based proteomics using Q Exactive, a high-performance benchtop quadrupole Orbitrap mass spectrometer. Molecular & Cellular Proteomics 2011, 10, M111. 011015.
- Lim, C.-K.; Lord, G. Current developments in LC-MS for pharmaceutical analysis. Biological and Pharmaceutical Bulletin 2002, 25, 547-557.
Reviewer 2 Report
In the scientific literature, the different myofibrillar protein fraction and the changes in myofibrillar proteins during postmortem aging of exudative pork longissimus thoracis meat was rarely investigated. In this manuscript, protein changes and differences in the myofibrillar protein fraction of pale, soft, and exudative and red, firm, and non-exudative pork longissimus thoracis were comparatively studied. The manuscript is interesting and properly written. The study is well planned with clear objectives and results are well represented. It fits with the scope and aims of the Journal.
Howewver, all abbreviations used in the figures and tables should be explained (eg. Tables 1).
Author Response
Dear reviewer
Thank you very much for the valuable comments concerning our manuscript. We have addressed all the comments as shown in the revised manuscript which we hope will now meet with your approval. Revised portions of the manuscript were tracked.
Reviewer 2:
1. In the scientific literature, the different myofibrillar protein fraction and the changes in myofibrillar proteins during postmortem aging of exudative pork longissimus thoracis meat was rarely investigated. In this manuscript, protein changes and differences in the myofibrillar protein fraction of pale, soft, and exudative and red, firm, and non-exudative pork longissimus thoracis were comparatively studied. The manuscript is interesting and properly written. The study is well planned with clear objectives and results are well represented. It fits with the scope and aims of the Journal.
However, all abbreviations used in the figures and tables should be explained (eg. Tables 1).
Response: Thank you very much for your kind advice. We have explained the abbreviations in each picture and table. We have added the explanation as follows: ˝RFN, red, firm, and non-exudative; PSE, pale, soft, and exudative; L*, lightness˝ in Table 1 (Line 240). Figure 1 (Line 261)., Table 2. (Line 265), Figure 2 (Line 278), Figure 3 (Line 317)., Table 3 (Line 321). and Figure 4 (Line 335). We have added the explanation as follows: ˝FC, fold change.˝ in Table 3 (Line 322).
Reviewer 3 Report
The manuscript presents interesting data regarding protein changes in PSE vs RFN pork. Comments and suggestions are attached.

Author Response
Dear reviewer:
Thank you very much for the valuable comments concerning our manuscript. We have addressed all the comments as shown in the revised manuscript which we hope will now meet with your approval. Revised portions of the manuscript were tracked.
Reviewer3
Title ´Comparative study on pale, soft and exudative (PSE) and red, firm and non-exudative (RFN) pork during postmortem aging: meat quality, protein changes and differential protein expression of myofibrillar fraction´ - from the title, one would expect that all listed traits i.e. ´meat quality, protein changes and differential protein expression of myofibrillar fraction´ will be analysed ´during postmortem aging´. However, this is not the case. According to the material and methods, results and the conclusion it seems that differential protein expression of myofibrillar fraction was made only 1h postmortem. The title should be better formulated to avoid misleading interpretation. Suggestion is to eliminate ´meat quality´ from the title while it seems that quality parameters (primarily pH) were used mainly to screen out samples. Also, there is no mention of meat quality in the aim and the conclusion. Otherwise, more parameters should be included in Table 1., like a*, b*, shear force etc. when talking about meat quality.
Response: Thank you very much and this is a good suggestion. We have eliminated ´meat quality´ in our title and we have modified the title as follow:
˝ Comparative study on pale, soft and exudative (PSE) and red, firm and non-exudative (RFN) pork: protein changes during aging and differential protein expression of myofibrillar fraction at 1 h postmortem ˝(Line 2-5)
Abstract:
- ´In this paper, the protein changes and differences in the myofibrillar protein fraction of pale, soft, and exudative (PSE) and red, firm, and non-exudative (RFN) pork longissimus thoracis were comparatively studied.´ – there is no mention of meat quality and postmortem aging - some of these analyses are done at different aging time
Response: This is a good suggestion. We have modified the sentence as follows: ˝ In this paper, the protein changes during aging and differences in the myofibrillar protein fraction at 1 d postmortem of pale, soft, and exudative (PSE) and red, firm, and non-exudative (RFN) pork longissimus thoracis were comparatively studied. ˝ (Line 15-17)
- ´PSE and RFN groups were screened out based on the differences in pH at 1 h, the L* and purge loss at 24 h postmortem.´ – How many samples? This is not clearly present also in material and methods. This sentence is confusing, samples were screened out according to pH at 1h and then those samples were screen out at 24h according to the L* and purge loss?!
Response: Thanks for your suggestion. Within 1 h postmortem, potential PSE and RFN meat were found as soon as possible at the slaughter line. Thus, the pH was measured from the 3rd rib to screen out possible PSE meat (eight carcass) and RFN meat (eight carcass) based on pH < 5.8 (possible PSE meat) or pH ≥ 5.8 (possible RFN meat) as the primary judgment [1,2]. After the eight PSE meat and eight RFN meat removed from the right half of each carcass, these samples were immediately removed from the carcass to determine of L* value, and weight at 1 h postmortem. After vacuum storage at 4℃ for 24 h, the pH, L* and purge loss were measured to confirm the PSE and RFN meat. Four PSE samples and four RFN samples were screened out according to the criteria developed by Chmiel et al.[1] and Warner et al.[2] with a slight modification: PSE, pH < 5.8, L* > 50 at 1 h postmortem, purge loss > 5% at 24 h postmortem; RFN, pH ≥ 5.8, L* ≤ 50 at 1 h postmortem, purge loss ≤ 5% at 24 h postmortem.
Thus, we have rewritten this paragraph (2.1. Sample collection) as follows: ˝A total of 16 castrated Duroc×Landrace×Yorkshire crossbred pigs with an average weight of 100 ± 10 kg were electrically stunned and slaughtered at a commercial meat processing plant (Su-shi Meat Co. Ltd., Jiangsu, Huai'an, China). The slaughtering process was carried out according to the Ordinance on Pig Slaughtering Management in China. The pH of LT was measured from the 3rd rib to screen out eight possible PSE meat (pH < 5.8) and eight possible RFN meat (5.8 < pH < 6.0) [1]within 1 h postmortem. The primarily screened longissimus thoracis (LT) between the 3rd and 13th rib were removed from the right half of each carcass. The LT muscle of each carcass was cut into three pieces with approximately equal size and weight. One piece of LT muscle was used for L* value detection at 1 h postmortem and then was minced to be frozen in liquid nitrogen and stored at −80 °C for further biochemical and proteomics analysis. Meanwhile, the other two pieces of meat were individually vacuum-packed and stored at 4°C for pH, L* and purge loss measurements at 24 h and 72 h postmortem aging. Meat samples for biochemical analysis at 24 h and 72 h postmortem were also prepared as same as above. Finally, four PSE samples and four RFN samples were screened out according to the criteria developed by Chmiel et al [1] and Warner et al [2] with a slight modification: PSE, pH < 5.8, L* > 50 at 1 h postmortem and purge loss > 5% at 24 h postmortem; RFN, pH ≥ 5.8, L* ≤ 50 at 1 h postmortem, purge loss ≤ 5% at 24 h postmortem. ˝(Line 70-87)
Introduction:
- ´Thus, this comparative study was conducted to identify the different myofibrillar protein fraction through proteomics and observe the changes in myofibrillar proteins during postmortem aging in PSE meat and RFN meat in order to gain a better understanding of the mechanism affecting meat quality.´ –If this is interpreted properly, the aim was not to identify the different myofibrillar protein fraction through proteomics during postmortem aging?! If so, make necessary changes to the title
Response: Yes, we have made changes to the title as indicated above and also this sentence as follows: ˝ Thus, this comparative study was conducted to identify the different myofibrillar protein fraction at 1 h postmortem through proteomics and observe the changes in myofibrillar proteins during postmortem aging in PSE meat and RFN meat in order to gain a better understanding of the mechanism affecting meat quality. ˝ (Line 63-67)
Materials and Methods
2.1. Sample collection
- ˝A batch of castrated Duroc x Landrace x Yorkshire pigs…˝ - specify exact number of pigs that were slaughtered
Response: We randomly selected 16 pigs at the commercial slaughter line. (Line 70)
- ˝The pH of LT was measured from the 3rd rib to screen out eight PSE and eight RFN meat˝ - specify pH value used to screen out PSE and RFN meat. Namely, mean pH value at 1h for RFN meat in Table 1. does not seem as the most representative for RFN meat. Please specify pH coefficient of the variation or standard deviation for RFN meat samples
Response: The classification of PSE and RFN was variable when only using pH as a judgment [3]. For example, pork loins were classified into PSE and RFN when the respective pH value were 5.32 and 5.64 [4]; 5.2 and 5.6 [5]; 5.47 and 5.69 [6]; and 5.60 and 5.96 [7]. In the current study, 5.8 < pH < 6.0 was set as the primary judgment for RFN meat samples according to the criteria developed by Chmiel et al.[1] and Warner et al.[2]. The PSE and RFN group were further verified by measurements of L* and purge loss at 24 h postmortem. As indicated above, PSE was selected based on pH < 5.8, L* > 50 at 1 h postmortem and purge loss > 5% at 24 h postmortem; RFN was selected based on pH ≥ 5.8, L* ≤ 50 at 1 h postmortem, purge loss ≤ 5% at 24 h postmortem(Line 84-87). The pH of RFN and PSE meat at 1 h postmortem was 5.88±0.33 and 5.48±0.16 , respectively as shown in Table 1 with a significant difference (P<0.05). (Line 239 Table 1.)
- Number of the analysed samples is not clear. Below Table 1., Table 2., and description of Figure 2 is written n=4. Eight were screen out, four analysed?! Four per aging group for all analyses or just proteomics?!
Response: As we described above, sixteen samples were initially screened out as the possible PSE (eight) and RFN (eight) meat according to the pH at 1 h, and then four PSE samples and four RFN samples were screened out according to the differences in pH and L* at 1 h, and purge loss at 24 h postmortem. Thus, four LT samples were used for each PSE and RFN group, which were for all parameters detection. (Line 74-75, Line 84-87)
- ˝The samples were vacuum-packed and stored at 4°C for quality measurements after 24 h and 72 h postmortem aging˝. - if all samples (not clear how many of them) were vacuum-packed at the same time, when and how was determined colour of the meat 1h postmortem. It was only mentioned that pH was measured to screen out PSE and RFN meat, but not that before vacuum packaging colour was measured. Was LT between 3rd and 13th rib cut down to parts and each individually vacuum-packed and aged for 24h and 72h ?!
Response: Firstly, we cut the entire LT sample into three equal parts. Then, one part was used to determine the color at 1 h postmortem, and the other two parts were individually vacuum-packed and stored at 4℃ and aged for 24 h and 72 h. There are enough members in our team to be responsible for each step, and the colorimeter measures quickly, that was why we accomplished. (Line 77-83)
- ´Meanwhile, approximately 50 g of the meat was minced and placed in the frozen storage tubes and snap-frozen in liquid nitrogen within 1 h postmortem, and stored at −80 °C for further biochemical and proteomics analysis.´ - for all biochemical and proteomics analyses were taken 1h?! What about analyses that were done on meat samples aged 24h and 72h postmortem?!
Response: Thanks for your advice. Samples for proteomics analysis were taken 1 h, and biochemical analyses were done on meat samples aged 1 h, 24 h and 72 h. Meat samples at 24 h and 72 h postmortem were prepared as same as the biochemical samples at 1 h postmortem. We have modified the description as above indicated. (Line 77-83)
- Suggestion is to change term ´purge loss´ to ´drip loss´ as this last one is used later in the discussion
Response: Thank you very much for your kind advice. Although both of them can be used as the parameters for water holding capacity, their determination methods are different. The purge loss samples are stored in a refrigerator at 4℃ after vacuum packaging [8], while drip loss samples are hung at 4℃ after vacuum packaging [9].
2.3. The myofibrillar fragmentation index The pH, color, and purge loss˝ – double title i.e. delete part of the title The pH, color, and purge loss
Response: Thank you very much for your kind advice. We have deleted the double title ˝The pH, color, and purge loss˝. (Line 104)
˝..then adjusted to 0.5 mg/mL to measure the absorbance at 540 nm.˝ - please specify device
Response: Thank you very much for your kind advice. We have added this information as follows: ˝(infinite 200 PRO multimode reader, Tecan, Switzerland)˝(Line 115-116)
- Results
3.1. Purge loss, pH, and color
- The LT were divided into the RFN meat group and PSE meat group according to the pH 1 h postmortem and L* and purge loss 24 h postmortem (Table 1).˝ - same as in the Abstract - this sentence is confusing; it seem that pH was determined only at 1h, and colour and drip loss at 24h postmortem. However, in Table 1. are presented pH and L for 1h and 24h. What happened with drip loss 72h? Shouldn´t it be - Samples were screen out according pH 1h and other quality parameters were determined for each group?!
Response: We are sorry for this misunderstanding.
Indeed, the pH and L* at 1 h and 24 h postmortem, and purge loss at 24 h and 72 h postmortem were measured. However, the pH and L* at 1 h postmortem and purge loss at 24 h postmortem was chosen to divide LT into RFN and PSE meat according to the method developed by Chmiel et al. [1] and Warner et al.[2] with a slight modification. Due to the limited time (within 1 h postmortem), we screened out the possible PSE and RFN meat according to pH at 1 h postmortem. After the selected meat removed from the carcasses at the slaughter line, these samples were immediately transferred to determine of pH, L*, and weight at 1 h postmortem. After vacuum storage at 4℃ for 24 h, the pH, L* and purge loss was measured. Then, we screened out ultimate PSE and RFN samples according to the differences in pH at 1 h, the L* and purge loss at 24 h postmortem. (Line 84-87)We also detected purge loss at 72h, it was not on display before since it’s not the selection criteria for the PSE sample. We have added purge loss at 72 h in Table 1. as follows: ˝RFN: 2.84±0.72; PSE: 8.12±3.46˝(Line 239 Table 1.)
- Suggestion is to add ˝postmortem˝ next to ˝Time (h)˝ in Table 1.
Response: Thank you very much for your kind advice. We have added ˝postmortem˝ next to ˝Time (h) ˝ in Table 1. (Line 239 Table 1.)
- As ´meat quality´ is listed in the title, it would be informative to present some other quality parameters of analysed samples, like a* and b*, shear force or to remove it from the title
Response: Thank you very much for your kind advice. We have removed ˝meat quality˝ from the title. (Line 2-5.)
- How do you explain increase in pH 24h for PSE meat?
Response: Thank you very much for your kind advice. The pH of PSE meat at 24 h postmortem was slight increase but not significant different from 1 h postmortem (P >0.05). Polak [10] also reported that pH of PSE meat remained unchanged during the aging period. It is speculated that the protein of LT muscle was degraded, the peptide fragment was broken during 24 of postmortem aging, and basic group of amino acids were released, resulting in a slight increase in pH [11].
- Again, n=4 below Table 1. is confusing – totally four samples or per each group were analysed?!
Response: We are sorry for the confusing. Four LT muscles for each PSE and RFN group were analyzed. (Line 266)
- Figure 1. b - post-mortal changes are marked with days (0d, 1d, 3d), however in the manuscript are written hours (0 h, 24h, 72h) – uniform marking is suggested
Response: Thank you very much for your kind advice. We have remarked Figure 1. (b) with hours ˝1 h, 24h, 72h˝. (Line 260 Figure 1.(b).)
- Figure 1. (a) Myofibrillar fragmentation index (MFI) value at 1 h, 24 h and 72 h postmortem of RFN and PSE meat; (b) Representative image of SDS-PAGE of the PSE and RFN samples during posmortem aging. a-b Values within the same aging time with different group are significantly different (p < 0.05). A-C Values within the same group with different aging time are significantly different (p < 0.05).˝ - there is no C label at a Figure 1.a, same for Figure 2.
Response: Thank you very much for your kind advice. We have revised ˝A-C˝ to ˝A-B˝ in Figure 1 and Figure 2. (Line 262 Figure 1. and Line 280 Figure 2.)
3.3. Desmin degradation
- ˝The western blotting detection of desmin is shown in Figure 3.˝ - please check figure number
Response: Thank you very much for your kind advice. We have revised ˝Figure 3.˝ to ˝Figure 2.˝
3.4. Protein identification and quantification – Suggestion is to add ´myofibrillar´ in this subtitle
Response: Thank you very much for your kind advice. We have added ´myofibrillar´ in this subtitle. (Line 281)
- ´Label-free Quantitative Proteomics was used to determine the significant difference in myofibrillar protein fraction 1 h postmortem between RFN and PSE meat groups. ´ - meat quality and protein changes were analysed during postmortem aging, LC-MS/MS analysis was done only 1 h postmortem? Would postmortem aging cause differences in myofibrillar protein fraction?! According to the title, one would expect that all listed traits (meat quality, protein changes and differential protein expression of myofibrillar fraction) were analysed ´during postmortem aging´. If this is not the case, the title needs to be better formulated
Response: Thanks for your suggestion. We have modified the title as indicated above. (Line 2-5) LC-MS/MS analysis was done at 1 h postmortem to detect the differential protein expression of myofibrillar fraction between PSE and RFN meat. It can be seen from detection of SDS-PAGE, desmin and MFI value that postmortem aging would cause differences in myofibrillar protein degradation. In order to avoid the influence of postmortem aging, the sample from 1 h postmortem was used to compare the protein expression differences of myofibrillar fraction between PSE and RFN meat.
- ´Figure 3. (a) LC/MS/MS identification of myofibrillar proteins shown in Venn diagram´ – diagram seems like surplus and suggestion is to omit from the manuscript
Response: Thank you very much for your kind advice. We have deleted the Venn diagram. (Line 316 Figure 3.)
- Figure 3. (b) Comparison of differential proteins between the PSE vs. RFN group shown in volcanic map – suggestion is to use term ´volcano plot´; would it be possible to compare number of up- and down-regulated differential myofibrillar proteins during different aging time?!
Response: Thank you very much for your kind advice. We have revised ˝volcanic map˝ to ˝volcano plot˝. (Line 318) The number of up- and down-regulated differential myofibrillar proteins were compared at 1 d postmortem and it is difficult to compare those difference during postmortem aging. As seen from the detection of SDS-PAGE, desmin and MFI, the proteins of myofibrillar fraction undergo intense postmortem changes, forming fragments or degraded polypeptide of proteins, which may confound with the intact protein for the quantification.
- Figure 3. (c) Heatmap with the difference FC > 1.5 or < 0.667 and a p < 0.05. –in the name of the heatmap PSE vs RFN should swap places i.e. while in the legend PSE is marked with turquoise blue and RFN with red it should be RFN vs PSE
Response: Thank you very much for your kind advice. We have switched ˝PSE˝ and ˝RFN˝ in the name of the heatmap. (Line 316 Figure 3. (d))
3.5. GO functional annotation of the differential proteins
- ´… complex subunit organization (Figure 4A).´ - change to small letter ´a´
Response: Thank you very much for your kind advice. We have revised ˝(Figure 4A)˝ to ˝(Figure 4(a))˝.(Line 327, Line 333)
- Text in the Figure 4. (a) and (b) has to small font and it is impossible to read it; suggestion is to put figures one below other and to try enlarge it
Response: Thank you very much for your kind advice. We have put figures one below other and enlarged the figures. (Line 334 Figure 4. (a)(b))
- Discussion
- ´The MFI of PSE meat was significantly higher than that of RFN meat (p < 0.05) (Figure 3),…´ - this is part of the results and there is no need to be in this form in the discussion
Response: Thank you very much for your kind advice. We have deleted this sentence and rewritten as follows: ˝Our result suggested that the myofibrils in PSE meat were prone to disruption.˝(Line 373-375)
-´This was consistent with the report from Allan et al.,…´ - this reference is missing in the list
Response: The reference number was 29, and now it is 32 (In 2.1. Sample collection, we have added two references). Since the reference number was marked at the end of the sentence, we have revised it to the end of the author's name now. In addition, we have revised ˝Allan˝ to ˝Wilhelm˝ according to the author´s name. (Line 375)
- ´The expression of HSP 90-alpha iso-form 2 and HSP beta-1 in PSE meat was significantly higher than that in RFN (p < 0.05).´ - this is part of the results and there is no need to be in this form in the discussion
Response: Thank you very much for your kind advice. We have rewritten this sentence as follows: ˝The expression of HSP 90-alpha isoform 2 and HSP beta-1 was found to be higher in PSE meat in the present study.˝(Line 416-418)
- ´Studies has found that apoptosis is related to the formation of PSE meat.´ - study or studies, please check
Response: Thank you very much for your kind advices. We have revised ˝Studies˝ to ˝Study˝. (Line 465)
- Conclusion
- ´In this study, the differences in the myofibrillar protein fraction of RFN and PSE meat 1 d postmortem and myofibrillar protein changes during 72 h postmortem aging were detected. ´ - this is inconsistent with the material and methods and the results. Is it 1h or 1d?!
Response: Thank you very much for your kind advice. We have revised ˝1 d˝ to ˝1 h˝. (Line 472)
- ´Moreover, a total of 172 proteins were identified 1 h postmortem as significant differential proteins between PSE and RFN meat,…´ - suggestion is to add ´1 h postmortem´
Response: Thank you very much for your kind advice. We have add ´1 h postmortem´in this sentence. (Line 477)
References
- Chmiel, M.; Słowiñski, M.; Janakowski, S. The quality evaluation of RFN and PSE pork longissimus lumborum muscle considering its microstructure. Annals of Animal Science 2014, 14, 737-747.
- Warner, R.; Kauffman, R.; Greaser, M. Muscle protein changes post mortem in relation to pork quality traits. Meat science 1997, 45, 339-352.
- Lesiów, T.; Xiong, Y.L. A simple, reliable and reproductive method to obtain experimental pale, soft and exudative (PSE) pork. Meat science 2013, 93, 489-494.
- Van Laack, R.L.; Kauffman, R.G. Glycolytic potential of red, soft, exudative pork longissimus muscle. Journal of Animal Science 1999, 77, 2971-2973.
- Lien, R.; Hunt, M.; Anderson, S.; Kropf, D.; Loughin, T.; Dikeman, M.; Velazco, J. Effects of endpoint temperature on the internal color of pork loin chops of different quality. Journal of food science 2002, 67, 1007-1010.
- Nam, K.; Ahn, D.; Du, M.; Jo, C. Lipid oxidation, color, volatiles, and sensory characteristics of aerobically packaged and irradiated pork with different ultimate pH. Journal of food science 2001, 66, 1225-1229.
- Kuo, C.; Chu, C. Quality characteristics of Chinese sausages made from PSE pork. Meat science 2003, 64, 441-449.
- Cardoso, G.P.; Dutra, M.P.; Fontes, P.R.; Ramos, A.d.L.S.; de Miranda Gomide, L.A.; Ramos, E.M. Selection of a chitosan gelatin-based edible coating for color preservation of beef in retail display. Meat science 2016, 114, 85-94.
- Gardner, M.A.; Huff-Lonergan, E.; Rowe, L.; Schultz-Kaster, C.; Lonergan, S.M. Influence of harvest processes on pork loin and ham quality. Journal of animal science 2006, 84, 178-184.
- Polak, T.; Došler, D.; Žlender, B.; Gašperlin, L. Heterocyclic amines in aged and thermally treated pork longissimus dorsi muscle of normal and PSE quality. LWT-Food Science and Technology 2009, 42, 504-513.
- Lawrie, R.A.; Ledward, D. Lawrie’s meat science; Woodhead Publishing: 2014.
Round 2
Reviewer 1 Report
After the reviewer's comments, the authors have improved the general scientific quality of the present manuscript, but it still needs revision.
Moreover, several typo are present in the text
As already said in the previous report, the authors have to keep in mind that other researchers could repeat the experiment and/or use it as a starting point for further investigations. Thus all the information related to the sample preparation and analytical methodology applied should be carefully reported.
Other comments
line 2020 -204: the sentence as it is is not clear and presents several typo, moreover "at nano-liter flow rate" is not clear if it is referred to the flow-rate injection or the flow-rate LC analysis
Line 177-178: the exhaustive abbreviation of nano liquid chromatography coupled to high-resolution nano-electrospray ionization tandem mass spectrometry is nLC-nESI-HRMS
Line 213: the explanation of Data Dependent Acquisition (DDA) is too general and should be better explained and supported by appropriate reference(s). The numbers of the parent ions selected from MS1, as well as their retention times and m/z values, as well as the cut-off of selection, should be reported.
Figure 3: The LC-MS chromatogram information is not enough. Please, be more specific (e.g. RP-LC-nESI(+)-MS chromatogram). Moreover, please specify if the LC-MS chromatogram is MS1 or MS2 (TIC or ion extracted chromatogram).
Please, be consistent in the analytical terminology (e.g. LC-MS/MS is ok, LC/MS/MS don't)
Author Response
Reviewer 1:
- After the reviewer's comments, the authors have improved the general scientific quality of the present manuscript, but it still needs revision.
Response: Thank you very much for your kind comments. We have revised this manuscript accordingly and we hope the changes made in the revised manuscript could meet with your approval.
- Moreover, several typo are present in the text
Response: Thank you very much for your kind advice. We have revised "frac-tion" to "fraction" (Line 5), "Cuisin" to "Cuisine" (Line 8), "are" to "is" (Line 51), "prelaughter" to "pre-slaughter" (Line 57), "were" to "was" (Line 144), "raw" to "row" (Line 248), "posmortem" to "postmortem" (Line 269), "thanks" to "thank" (Line 517). Besides, the repeated "." has been deleted. (Line 87).
- As already said in the previous report, the authors have to keep in mind that other researchers could repeat the experiment and/or use it as a starting point for further investigations. Thus all the information related to the sample preparation and analytical methodology applied should be carefully reported.
Response: Thank you very much for your kind advice. As you suggested, we have carefully revised the paragraph of "2.5. LC-MS/MS analysis" and made minor changes of the other method description for a better understanding of our investigation (Line 157-217).
Other comments
- line 202 -204: the sentence as it is is not clear and presents several typo, moreover "at nano-liter flow rate" is not clear if it is referred to the flow-rate injection or the flow-rate LC analysis
Response: Thank you very much for your kind advice. We have revised the sentence as follows: "The 20 precursor ions with the highest intensity from each MS1 scan were selected for high energy collision dissociation (HCD). A resolution of 15,000 (m/z: 200), an AGC target of 1 × 105, a maximum IT of 50 ms, an intensity threshold of 2 × 104, an isolation window of 1.6 m/z, and a normalized collision energy of 28 were set in MS2 scan. The number of microscans was set as 1 and the scanned peptides were dynamically excluded for 30 s. "(Line 157-217). The sentence "at nano-liter flow rate" is referred to the flow-rate of LC-MS/MS analysis (Line 187).
- Line 177-178: the exhaustive abbreviation of nano liquid chromatography coupled to high-resolution nano-electrospray ionization tandem mass spectrometry is nLC-nESI-HRMS
Response: Thank you very much for your kind advice. We have revised ˝nano LC-MS/MS˝ to ˝nLC-nESI-HRMS˝.(Line 159)
- Line 213: the explanation of Data Dependent Acquisition (DDA) is too general and should be better explained and supported by appropriate reference(s). The numbers of the parent ions selected from MS1, as well as their retention times and m/z values, as well as the cut-off of selection, should be reported.
Response: Thank you very much for your kind advice. We have rewritten the sentence as follows: ˝DDA is an accelerated and autonomous data acquisition mode [1]. In this mode, the MS instrument performed MS full-scan (MS1 scan) immediately followed by MS2 analysis on a list of precursor ions selected by their abundances from the full-scan spectrum [1,2].˝(Line 198-201). We have also reported the parameters as follows: ˝MS1 scan was performed and parameters were set as follows: scanning range of parent ion, 300-1800 m/z; resolution, 60,000 (m/z: 200); automatic gain control (AGC) target, 3 × 106; maximum injection time (IT), 50 ms. The number of microscans was set as 1.˝(Line 202-205)
- Figure 3: The LC-MS chromatogram information is not enough. Please, be more specific (e.g. RP-LC-nESI(+)-MS chromatogram). Moreover, please specify if the LC-MS chromatogram is MS1 or MS2 (TIC or ion extracted chromatogram).
Response: Thank you very much for your kind advice. We have revised the sentence as follows: ˝Figure 3. (a) Representatively mass spectrometry (MS) base peak chromatogram of myofibrillar protein extracted from RFN meat by nLC-nESI-HRMS; (b) MS base peak chromatogram of myofibrillar protein extracted from PSE meat by nLC-nESI-HRMS;˝ (Line 323-325). The high peaks and strong signals in the figure indicate that the proteins in samples were complex and abundant with expected degree of separation.
The Figure 3. is the base peak chromatogram. The signal of all precursor ions in one full scan presents as the mass spectrum of MS1. The mass spectrum of MS2 is the plot of the fragment ions signal dissociated from the 20 precursor ions with the highest intensity. The mass spectrum in Figure 3 is essentially the combination of one full scan (MS1) and 20 MS2 plots. The base peak chromatogram is similar to the TIC chromatogram, however, it monitors the most intense peak in each spectrum [3]. The total ion current (TIC) chromatogram represents the summed intensity across the entire range of masses being detected at every point in the analysis [3]. The range is typically several hundred mass-to-charge units or more. In complex samples, the TIC chromatogram often provides limited information as multiple analytes elute simultaneously, obscuring individual species (https://en.wikipedia.org/wiki/Mass_chromatogram). The base peak chromatogram represents the intensity of the most intense peak at every point in the analysis. Base peak chromatogram is more informative than TIC chromatograms because the background is reduced by focusing on a single analyte at every point (https://en.wikipedia.org/wiki/Mass_chromatogram).
- Please, be consistent in the analytical terminology (e.g. LC-MS/MS is ok, LC/MS/MS don't)
Response: Thank you very much for your kind advice. We have been consistent in the analytical terminology.
References
- Guo, J.; Huan, T. Comparison of Full-Scan, Data-Dependent, and Data-Independent Acquisition Modes in Liquid Chromatography–Mass Spectrometry Based Untargeted Metabolomics. Analytical Chemistry 2020, 92, 8072-8080, doi:10.1021/acs.analchem.9b05135.
- Fernández-Costa, C.; Martínez-Bartolomé, S.; McClatchy, D.B.; Saviola, A.J.; Yu, N.-K.; Yates, J.R. Impact of the Identification Strategy on the Reproducibility of the DDA and DIA Results. Journal of Proteome Research 2020, 19, 3153-3161, doi:10.1021/acs.jproteome.0c00153.
- Murray, K.K.; Boyd, R.K.; Eberlin, M.N.; Langley, G.J.; Li, L.; Naito, Y. Definitions of terms relating to mass spectrometry (IUPAC Recommendations 2013). Pure and Applied Chemistry 2013, 85, 1515-1609.